# Effects of Condensed Tannins on Bacterial and Fungal Communities during Aerobic Exposure of Sainfoin Silage

**DOI:** 10.3390/plants12162967

**Published:** 2023-08-17

**Authors:** Rongzheng Huang, Chunhui Ma, Fanfan Zhang, Xuzhe Wang

**Affiliations:** Grassland Science, School of Animal Technology, Shihezi University, Shihezi 832000, China; huangrz2013@163.com (R.H.); wangxuzhe123@sohu.com (X.W.)

**Keywords:** sainfoin, silage, condensed tannins, aerobic exposure, bacteria, fungi

## Abstract

Background: Sainfoin is a forage legume that is widely distributed around the world and is beneficial for animals owing to the characteristics of its condensed tannins (CTs), which, from certain plants, can prolong the aerobic stability of silage. Methods: The present study investigated whether sainfoin CTs can prolong aerobic stability by adding polyethylene glycol (PEG) to inactivate CT activity in the silage system. Results: The results showed that aerobic stability increased under the PEG treatment (*p* < 0.05). Ammonia nitrogen (0.71 g/kg DM vs. 0.94 g/kg DM; *p* < 0.05) was higher in the PEG-treated group compared with the control after 3 d of aerobic exposure. BA was detected only in the PEG-treated group upon aerobic exposure. Yeasts were more abundant in the control compared with the PEG-treated group after 7 d of aerobic exposure, after which the relative abundance of *Lactobacillus* was lower in the PEG-treated group (65.01% vs. 75.01% in the control; *p* < 0.05), while the relative abundance of *Pediococcus* was higher in the PEG-treated group compared with the control (10.9% vs. 4.49%, respectively; *p* < 0.05).The relative abundances of *Apiotrichum* and *Aspergillus* were lower in the control than in the PEG-treated group after 7 d of aerobic exposure. Conclusions: The results suggested that sainfoin CTs decreased aerobic stability, but could inhibit certain bacteria and fungi, such as *Pediococcus* and *Apiotrichum*, and preserve the protein content during the aerobic exposure of silage.

## 1. Introduction

Sainfoin (*Onobrychis viciifolia*) is a forage legume that is widely distributed in Europe, Asia, and North America [1]. Sainfoin has several benefits for the agricultural industry, including its drought tolerance and nitrogen-fixing activity, protein protection in the rumen, methane reduction, and the inhibition of rumen protozoa [2]. The condensed tannins (CTs) from sainfoin are primarily responsible for these effects in ruminants, which are due to the formation of CTs–protein complexes and the stabilization of CTs–protein complexes in certain environments [3]. Many legumes contain CTs, but sainfoin is considered superior because it has the highest capacity to bind protein and causes the least inhibition of cellulose digestion by rumen bacteria [4]. Consequently, sainfoin could be a potential high-quality forage for animals. In the production of animal feed, haymaking and ensiling are common methods used to preserve forage. Among these, ensiling is an important method used to preserve forage [5]. It is well known that the fermentation of forage under anaerobic environments through lactic acid bacteria (LAB) results in the production of organic acid and a decreased pH, which extends the associated feed storage time and improves the feed palatability [6]. During ensiling, changes in the microbial community play a key role in fermentation. Numerous studies have focused on the ensiling of crops, such as corn, alfalfa, barley, and ryegrass, to clarify the microbial community during ensiling through the use of next-generation sequencing techniques (NGS) [7].

The use of legume silage, such as alfalfa, usually results in protein degradation due to its higher protein content and buffering capacity compared with grass silage [8]. Several studies have observed that protein degradation is inhibited by CTs during ensiling [9,10]. This effect still appears after the aerobic exposure of silage, and the aerobic stability is prolonged due to the presence of existing with CTs [9]. However, the addition of CTs from the quebracho plant (*Schinopsis lorentzii*) showed no effect on the aerobic stability of corn silage, but decreased the aerobic stability when the addition level exceeded 50 g CTs/kg dry matter (DM) [11]. The differences between the findings of the two studies probably occurred because the effect of CTs, which are obtained from different plants, on silage systems is complex [3]. In general, CTs can inhibit some bacteria and fungi [9,12]. Recently, our previous study observed that CTs from sainfoin could inhibit protein degradation via the inhibition of protease and *Pediococcus* activity, and inhibited some yeast during the early stage of ensiling [13,14]. In general, yeast can initiate aerobic deterioration after silage is exposed to the air. Thus, whether the effects of CTs on fermentation and the microbial community, especially that of yeast, during the aerobic exposure of sainfoin silage have the capacity to prolong aerobic stability needs further investigation. Therefore, the present study was conducted to investigate the effects of CTs on the bacterial and fungal community via NGS techniques and the fermentation characteristics during aerobic exposure of sainfoin silage.

## 2. Results

### 2.1. Characteristics of Fermentation during Aerobic Exposure in Sainfoin Silage

The silage characteristics are shown in Table 1. The pH was highest in the control compared with PEG treatment at 7 d of aerobic exposure (*p* < 0.05). The water-soluble carbohydrate (WSC) content was lower in the polyethylene glycol (PEG) treatment after 60 d of ensiling (*p* < 0.05), but there was no significant difference in the WSC between the control and the PEG-treated group at 7 d of aerobic exposure (*p* > 0.05). The lactic acid (LA) and acetic acid (AA) contents decreased in both the control and PEG-treated groups after 3 d of aerobic exposure (*p* < 0.05), but the LA and AA contents showed no difference between the two groups (*p* > 0.05). BA was detected in the PEG-treated group but was not detected in the control group after 60 d of ensiling and 7 d of aerobic exposure. The yeast count was lower in the control compared with the PEG-treated group after 60 d of ensiling (*p* < 0.05) but higher in the control compared with the PEG-treated group after 7 d of aerobic exposure (*p* < 0.05). The mold count showed the same results as the yeast count. The PEG-treated group showed higher aerobic stability than the control (182 h vs. 156 h, respectively, *p* < 0.05).

### 2.2. Bacterial Community during Aerobic Exposure

The alpha diversity of bacteria after 60 d of ensiling and aerobic exposure in sainfoin silage is shown in Table 2. The Good’s coverage (>0.999) results indicated that the degree of sequencing was sufficient for the bacterial community analysis. The results revealed that there was no significant difference in the richness and diversity of the bacteria community between the control and the PEG-treated group during 7 d of aerobic exposure (*p* > 0.05). The beta analysis of the bacterial community showed that there was no significant difference between the control and the PEG-treated group during 7 d of aerobic exposure (Figure 1, R^2^ = 0.0810, *p* = 0.0560).

At the phylum level, as shown in Figure 2a, Firmicutes was dominant during 7 d of aerobic exposure, followed by Actinobacteria and Proteobacteria, with relative abundances of 80.81–96.73%, 1.28–6.60%, and 0.93–2.63%, respectively. There was no difference in the relative abundances of these bacteria between the control and the PEG-treated group after 60 d of ensilage and 7 d of aerobic exposure (*p* > 0.05). After 3 d of aerobic exposure, the relative abundance of Firmicutes in the PEG treatment was higher than that in the control (96.46 vs. 93.97%; *p* < 0.05), but the relative abundance of Proteobacteria was lower than that of the control (1.01 vs. 2.27%; *p* < 0.05).

At the genus level, as shown in Figure 2b, *Lactobacillus* was dominant during 7 d of aerobic exposure, followed by *Weissella*, *Pediococcus*, and *Rhodococcus*, with relative abundances of 60.36–78.42%, 9.98–18.44%, 3.42–12.53%, and 1.22–3.35%, respectively. After 60 d of ensiling, the relative abundances of these four bacteria showed no differences (*p* > 0.05). The same results were observed after 3 d of aerobic exposure (*p* > 0.05). After 7 d of aerobic exposure, the relative abundance of *Lactobacillus* was lower in the PEG-treated group than in the control (65.01 vs. 75.01%, respectively; *p* < 0.05), while the relative abundance of *Pediococcus* was higher in the PEG-treated group when compared with the control (10.9 vs. 4.49%, respectively; *p* < 0.05). Additionally, the relative abundance of *Enterobacter* in the control was higher than that in the PEG-treated group at 7 d of aerobic exposure (1.54–2.3 vs. 0.46–0.56%; *p* < 0.05).

### 2.3. Fungal Community during Aerobic Exposure

The alpha diversity of fungi after 60 d of ensiling and aerobic exposure in sainfoin silage is shown in Table 3. The Good’s coverage results (>0.999) indicated that the degree of sequencing was sufficient for the fungal community analysis. The results showed that there was no difference in the diversity of the fungal community between the control and the PEG-treated group during 7 d of aerobic exposure (*p* > 0.05). Similarly, the beta analysis of the fungal community indicated that there was no clear difference between the control and the PEG-treated group during 7 d of aerobic exposure (Figure 3, R^2^ = 0.4715, *p* = 0.001).

At the phylum level, as shown in Figure 4a, Ascomycota (relative abundance of 48.47–77.76%) was dominant in both the control and PEG-treated silage groups after 60 d of ensiling and during 7 d of aerobic exposure.

At the genus level, as shown in Figure 4b, *Cladosporium* was dominant in both the control and the PEG-treated group after 60 d of ensiling and 3 d of aerobic exposure, followed by *Alternaria* and *Vishniacozyma*. After 3 d of aerobic exposure, the relative abundances of *Alternaria* and *Filobasidium* were both higher in the control than in the PEG-treated group (23.02 vs. 6.88% for *Alternaria* and 6.4 vs. 1.79% for *Filobasidium*; *p* < 0.05). After 7 d of aerobic exposure, the relative abundances of *Aspergillus* and *Apiotrichum* (0.58 vs. 7.19% for *Aspergillus* and 2.28 vs. 8.63% for *Apiotrhicum* in the control vs. the PEG-treated group, respectively) were lower but that of *Stemphylium* (3.00 vs. 0.78%) was higher in the control than in the PEG-treated group (*p* < 0.05).

## 3. Discussion

### 3.1. Aerobic Stability and Fermentation

The results showed that the aerobic stability of the PEG-treated group was higher than that of the control. In general, the DM, AA, and BA contents and the yeast and mold counts are the most important factors affecting silage aerobic stability [15]. DM usually has a negative effect on temperature rise during aerobic exposure, in which the temperature of crops with higher DM contents (300–500 g/kg fresh weight) rises more rapidly compared to crops with lower DM contents (150–300 g/kg fresh weight) [8]. According to the results of the present study, the DM content was 247.88–264.4 g/kg in both the control and PEG-treated group during aerobic exposure, which indicated that in this situation, the DM had no effect on aerobic stability. The contents of LA and AA were both lower when combined with CTs from purple prairie clover (PPC, *Dalea purpurea*), but the aerobic stability was prolonged during 14 d of aerobic exposure to silage, indicating that CTs directly inhibited yeast activity [9]. Consequently, the CTs from PPC had a strong capacity to prolong aerobic stability. However, CTs from quebracho had no effect on the aerobic stability of silage, although these CTs had the same effects on LA and AA contents as the CTs from PPC during the aerobic exposure of silage [11]. Thus, CTs and organic acids could be the two most important factors affecting aerobic stability, and the difference between the two silages could be attributed to the different CTs from different materials. The present study showed that CTs could decrease aerobic stability. In addition, the initial organic acid content and water activity determine the relationship between the residual WSC and the aerobic stability of silage [16]. The residual WSC can not only provide the substrate for yeast but can also result in the reduction of water activity [15]. The present study showed that the content of WSC exhibited no difference between the control and the PEG-treated group (*p* > 0.05), which indicated that the organic acid content was probably the main factor influencing the aerobic stability of silage.

There were two peaks of silage temperature, with the yeast and aerobic AA bacteria causing the first peak, followed by mold development [17]. It is well known that yeasts grow rapidly during the aerobic exposure of silage, thereby initiating the aerobic deterioration process. Previous studies have shown that the inhibitory effect of CTs on yeast occurs during ensiling [13]. However, the present study showed that the count of yeast was 13.67% higher in the control than in the PEG-treated group after 7 d of aerobic exposure. The results suggest that CTs from sainfoin lost their ability to inhibit yeasts during the aerobic exposure of silage.

Fungi can usually be inhibited by undissociated short-chain fatty acids during aerobic exposure to silage [15]. Theoretically, LA should be more dissociated than other short-chain fatty acids because it has the lowest p*K*a compared with PA and AA [18], that is, it results in rapid, decreased pH but lowers the capacity to inhibit fungal growth. PA has shown the highest capacity to inhibit yeasts and molds, followed by AA and LA [19]. Thus, researchers have observed that well-fermented silage that contains higher levels of LA among the total fermentation acids is prone to instability during aerobic exposure [20,21]. Furthermore, the high positive correlation between the concentration of AA and aerobic stability suggests that AA can improve the aerobic stability of silage [22]. AA is an antifungal agent, and a decreased AA content results in aerobic deterioration during the aerobic exposure of silage [23]. The results of the present study showed that the AA content did not differ between the control and the PEG-treated group after 7 d of aerobic exposure. Thus, the AA content was not a factor in the increased yeast in the control after 7 d of aerobic exposure. BA has also been found to increase the aerobic stability of silage [24,25,26]. The results of the present study showed that the BA content was stabilized in the PEG-treated group after 60 d of ensiling and 7 d of aerobic exposure, while no BA was detected in the control. A meta-analysis showed that CTs had a dose-dependent effect on the inhibition of BA production [5]. The results of the present study suggested that the higher aerobic stability of the PEG-treated group was due to the BA content, while the CTs from sainfoin probably inhibited BA production, which indirectly caused the yeast to grow faster upon the aerobic exposure of sainfoin silage.

The crude protein content decreased by 5.93% with the addition of PEG during the aerobic exposure of silage. Consequently, the AN content increased to 24.46% in the PEG-treated group compared with the control after 3 d of aerobic exposure. Microbial activity, such as the activity of *Entrobacter* and *Clostridium*, is the main contributor to AN production during ensiling, and the activity of both *Entrobacter* and *Clostridium* is inhibited at pH values below 4.5 [8]. The results of the present study showed that the pH value was above 4.5 in both the control and PEG-treated group during 7 d of aerobic exposure, indicating that CTs could be the main reason for this difference. The formation of protein–tannin complexes composed of sainfoin CTs and lucerne fraction 1 protein was found to be stable in the pH range of 4.0–6.5 [27]. The findings of the present study suggested that CTs inhibited protein degradation during the aerobic exposure of silage.

### 3.2. Bacterial Community

At the phylum level, Firmicutes was dominant in both the control and the PEG-treated group during 7 d of aerobic exposure. Several studies have observed that the dominant bacteria at the phylum level shifted from Firmicutes to Proteobacteria after aerobic exposure to silage. The results of the present study suggested that silage quality probably stabilized during 7 d of aerobic exposure. The addition of PEG decreased Proteobacteria but increased Firmicutes when compared with the control after 3 d of aerobic exposure. The increased Firmicutes activity following the addition of PEG was likely related to the decreased pH [28]. This impact should have persisted after 7 d of aerobic exposure as the pH of the PEG-treated group was the lowest. However, the impact of PEG on Firmicutes disappeared when the aerobic exposure time was prolonged (7 d). The present results suggested that the change in bacterial community composition was probably complicated due to the presence of CTs during the aerobic exposure of sainfoin silage.

At the genus level, the bacterial community of the PEG-treated group was mainly consistent with that of the control after 60 d of ensiling. The same results were observed after 3 d of aerobic exposure, except for the finding that *Enterobacter* showed higher activity in the control compared with the PEG-treated group. Consistently, *Enterobacter* still showed higher activity in the control after 7 d of aerobic exposure. *Enterobacter* shows less activity at pH values below 4.5 [29]. In the present study, the pH was over 4.5 in the control group during 7 d of aerobic exposure. CTs from sainfoin have shown the ability to inhibit some strains of *Enterobacter* [12]. However, combined with the findings of our previous study, this ability probably disappeared in the silage system in both the anaerobic fermentation and aerobic exposure stages [13]. Thus, the pH could be the main factor stimulating *Enterobacter* activity.

*Lactobacillus* was dominant in both the control and the PEG-treated group during 7 d of aerobic exposure. Similar results were observed in corn and wheat silage after exposure to the air [21,30]. After 7 d of aerobic exposure, the *Lactobacillus* activity in the PEG-treated group was lower than that of the control, while the *Pediococcus* activity exhibited the opposite trend. The pH was 4.5–4.6 in both the control and the PEG-treated group during 7 d of aerobic exposure, and the pH in the PEG-treated group was lower compared with control. *Pediococcus* can grow well under pH values ranging from 4.3–4.9 during ensiling [31]. The pH, in the present study, could not influence *Pediococcus* activity during 7 d of aerobic exposure. In our previous study, CTs from sainfoin showed the ability to inhibit *Pediococcus* but had no impact on *Lactobacillus* during ensiling [13]. Additionally, there were antagonistic effects between *Lactobacillus* and *Pediococcus* during ensiling [14]. The results of the present study suggested that *Lactobacillus* activity increased due to the indirect effect of the inhibition of *Pediococcus* activity by CTs during the aerobic exposure of silage.

Notably, the PEG-treated group contained BA during the aerobic exposure of silage, but only some *Clostridium* genera were detected, such as *Clostridium_sensu_stricto_1*. The relative abundance of this bacteria did not differ between the control and the PEG-treated group (*p* > 0.05), and the relative abundance of this bacteria was far below 1%. Other research did not detect BA during sainfoin ensiling, and the relative abundance of *Clostridium tyrobutricum* was 27.89% [32]. The results suggested that the CTs from sainfoin inhibited BA production, but the mechanism needs further study.

### 3.3. Fungal Community

The fungal richness decreased during aerobic exposure, in accordance with the findings of previous research [28]. High fungal diversity can improve the aerobic stability of whole-plant corn silage [33]. In the present study, the fungal diversity showed no difference between treatments or between days of aerobic exposure (*p* > 0.05), indicating that the CTs did not prolong the aerobic stability of sainfoin silage. The fungal community composition was the same between the control and the PEG-treated group during 7 d of aerobic exposure (Figure 4). The most abundant fungal genera during 7 d of aerobic exposure were *Cladosporium*, *Alternaria*, *Wickerhamomyces*, *Vishniacozyma*, and *Apiotrichum*, among which *Wickerhamomyces*, *Vishniacozyma*, and *Apiotrichum* belonged to the yeasts.

*Wickerhamomyces* showed overall competitiveness due to its high tolerance for stressful environments, such as low pH levels, water activity, and the presence of LAB, as well as its ability to metabolize a large number of carbon and nitrogen sources and produce toxins [34]. The relative abundance of *Wickerhamomyces* was below 1% after 3 d but increased rapidly, resulting in dominance after 7 d of aerobic exposure in both the control and PEG-treated group. Additionally, some strains of *Wickerhamomyces* have shown the ability to produce substantial amounts of AA through the fermentation of WSC [35]. The present findings suggested that, although the growth of *Wickerhamomyces* probably slowed, it could become the predominant fungi in the silage system upon aerobic exposure. Notably, CTs showed no effect on *Wickerhamomyces* activity during the aerobic exposure of silage.

*Apiotrichum* is an anamorphic basidiomycetous yeast genus that is widely distributed around the world [36]. Some strains of *Apiotrichum* have shown good AA tolerance and assimilation capability, and grow well in different ratios of AA-, PA-, and BA-based mixtures through the use of these acids as substrates to produce lipids [37]. This capability remains functional in silage systems. Researchers observed that *Apiotrichum* was the dominant fungi during ensiling when the AA content was the highest (2.48% DM) when compared with other treatments [38]. In the present study, the activity of *Apiotrichum* in the PEG-treated group was higher than that of the control after 7 d of aerobic exposure (the relative abundance of *Apiotrichum* was 8.63 vs. 2.28%). The present results showed that the AA content was below 1.65% DM and showed no difference between the control and the PEG-treated group after 7 d of aerobic exposure, indicating that AA probably did not affect *Apiotrichum* activity. Thus, the results probably suggested that CTs had a strong ability to inhibit *Apiotrichum* growth in the silage system.

## 4. Materials and Methods

### 4.1. Silage Preparation

Sainfoin (*Onobrychis viciifolia* socp. cv. Qi-Tai, BY2020-003, grassland farming, Xinjiang, China) seeds were sown in an experimental field of Shihezi University on 1 May 2022 (N 44.21; E 85.57, Xinjiang, China). The study area has a temperate desert climate and the average annual rainfall is 128 mm. Whole sainfoin plants were harvested in July 2022 at the early flower stage. Samples were wilted until DM content approximated 250 g/kg fresh weight, then the plants were chopped into 1–2 cm pieces. Polyethylene glycol (PEG) (Sigma-Aldrich, Shanghai, China; molecular weight, 6000) was used to inactivate CT activity (the ratio of PEG:CT was 2:1, based on DM). The PEG water solution (640 g/L) was prepared and sprayed on samples, and the CT concentration was approximately 50 g/kg DM [13]. After treatment, 1000 g samples were packed into polyethylene plastic bags (30 cm × 50 cm), then compacted and sealed using a vacuum sealer. Five replicates were prepared for the control and PEG treatment. All bags were stored indoors at 20 °C.

### 4.2. Aerobic Exposure of Silage

After 60 d of fermentation, 3 kg samples of each treatment were transferred to plastic jars (5 L volume, diameter 20.40 cm, height 15.3 cm). Each jar was covered with five layers of cheesecloth to avoid the introduction of impurities and stored at 20 °C for 8 d. A temperature monitoring instrument (i500-E8T, Yuhuan Zhituo Instrument Technology Co., Ltd., Hangzhou, China) was used to record the internal temperature (the temperature probes were embedded in the middle layers of silage) and the ambient temperature (temperature probes were placed outside of the jar) of the silage every 15 min during 8 d of aerobic exposure. The silage was spoiled when the internal temperature was 2 °C higher than the ambient temperature [39].

Samples were taken from each treatment after 60 d of ensiling and at 3 and 7 d of the aerobic exposure of sainfoin silage. The samples were stored at −20 °C and −80 °C for the determination of their characteristics and microbial community analysis, respectively.

### 4.3. Characteristics Analysis of Silage

Two-hundred-gram silage samples were dried at 65 °C for 48 h and ground to obtain the DM content through the use of a 1.00 mm griddle. The water-soluble carbohydrate content was determined via extract samples through water solution at 100 °C for 30 min, then analysis was conducted according to the anthrone method [40]. An automatic Kjeldahl nitrogen analyzer (K9840, Hanon Co., Ltd., Qingdao, China) was used to determine the nitrogen content of samples according to the guidelines of the Association of Official Agricultural Chemists.

To determine the fermentation characteristics, 20 g silage samples were combined with distilled water and thoroughly blended in a homogenizer (L-1BA, Kuansons Biotechnology Co., Ltd., Shanghai, China). After filtering the mixture, the supernatant was used for the analysis of organic acids and ammonia nitrogen (AN). The AN was determined according to the phenol-hypochlorite colorimetric methods [41]. The lactic acid (LA), acetic acid (AA), propionic acid (PA), and butyric acid (BA) were analyzed following the methods [13].

For the microbial count, 10 g samples were homogenized in 90 mL sterilized saline (0.8%) and the supernatant was serially diluted (10^2^–10^8^). The microbial incubation was accomplished according to the methods described in a previous study [42]. Yeasts and molds were counted after incubation on Rose–Bengal agar at 25 °C for 78–120 h. LABs were counted after incubation on de Man–Rogosa–Sharpe (MRS) agar at 30 °C for 24 h. Aerobic bacteria were counted after incubation on nutrient agar at 30 °C for 24 h. The number of colony-forming units (CFUs) was expressed per gram of fresh forage, and the microbial count was presented as log-transformed before statistical analysis.

### 4.4. DNA Extraction and Sequence Analysis of Bacteria and Fungi

The total DNA of each sample was extracted with a commercial DNA Kit (FastDNA^®^ Spin Kit for Soil, MP Biomedicals, New York, NY, USA). Primers targeted the V3–V4 (338F: ACTCCTACGGGAGGCAGCAG; 806R: GGACTACHVGGGTWTCTAAT) regions of bacterial 16S rDNA and the ITS1 regions (ITS1F: 5′-CTTGGTCATTTAGAGGAAGTAA-3′; ITS2R: 5′-GCTGCGTTCTTCATCGATGC-3′) of fungi. The amplicons were extracted, purified, and analyzed according to methods described in [43]. The Quantitative Insights Into Microbial Ecology (QIIME, v1.8.0) pipeline was employed to process the sequencing data according to methods described in [44]. The operational taxonomic unit classification was conducted by using BLAST to search the representative sequences set against the Greengenes Database using the best hit method [45]. The raw sequences of all samples were deposited in the National Center for Biotechnology (NCBI) Sequence Read Archive (SRA) under accession numbers “PRJNA948115” and “PRJNA953494” for bacteria and fungi, respectively (accessed on 1 June 2023).

### 4.5. Statistical Analysis

The characteristics of sainfoin silage were subjected to a two-way analysis of variance, 2 × 3 factorial complete randomized design (PEG and control groups × 60 fermentation d, and aerobic exposure for 3 and 7 d). Data were analyzed using IBM SPSS 22 Statistics (IBM Corp., Armonk, NY, USA). Significant differences between treatments were determined using Tukey’s test at *p* < 0.05.

## 5. Conclusions

In this study, CTs inhibited BA production in the silage system, and the PEG treatment increased the aerobic stability of silage due to BA production. The crude protein content was stabilized and the AN decreased with the presence of CTs. *Lactobacillus* was dominant during 7 d of the aerobic exposure to silage. *Wickerhamomyces* grew slowly but would become dominant with prolonged aerobic exposure time, and its activity was not affected by CTs. CTs could inhibit some bacteria and fungi, such as *Pediococcus* and *Apiotrichum*, during the aerobic exposure of silage, which probably had no effect on the aerobic stability of silage.

## Figures and Tables

**Figure 1 plants-12-02967-f001:**
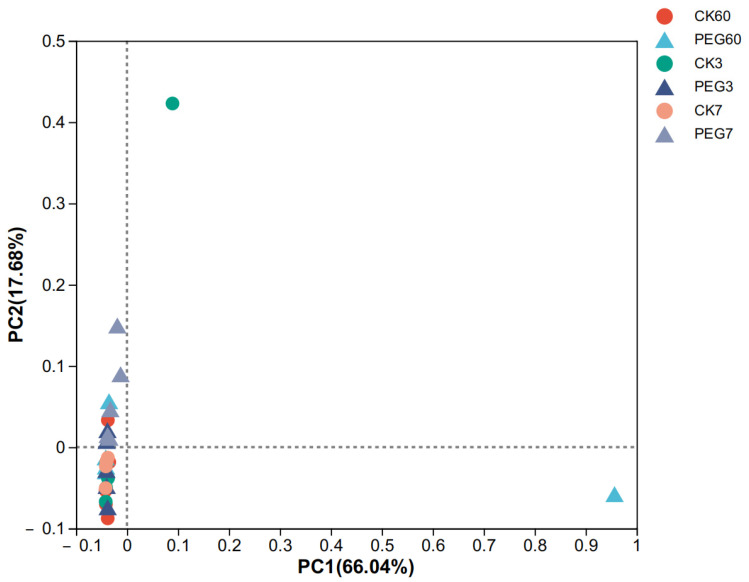
Principal coordinate analysis (PCoA) plot based on the weighted UniFrac distance for the bacterial community of sainfoin silage. CK60: sainfoin ensiling after 60 d. CK3: sainfoin silage after 3 d of aerobic exposure. CK7: sainfoin silage after 7 d of aerobic exposure. CK: control; PEG: polyethylene glycol (R^2^ = 0.0810, *p* = 0.0560).

**Figure 2 plants-12-02967-f002:**
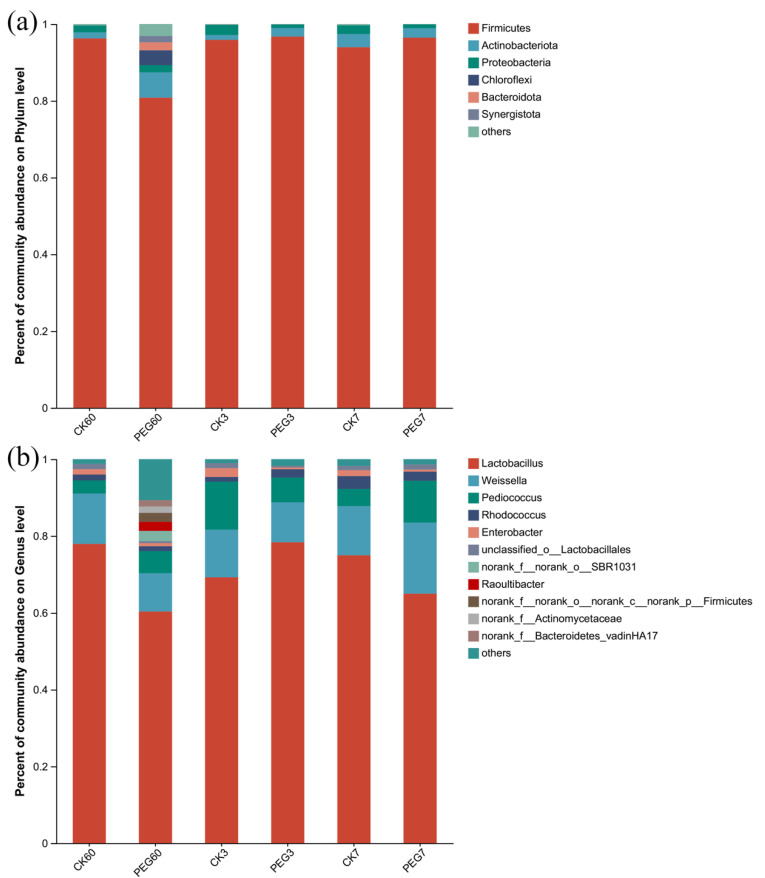
Bacterial community at the (**a**) phylum and (**b**) genus level during the aerobic exposure of sainfoin silage. CK60: sainfoin ensiling after 60 d. CK3: sainfoin silage after 3 d of exposure. CK7: sainfoin silage after 7 d of exposure. CK: control; PEG: polyethylene glycol.

**Figure 3 plants-12-02967-f003:**
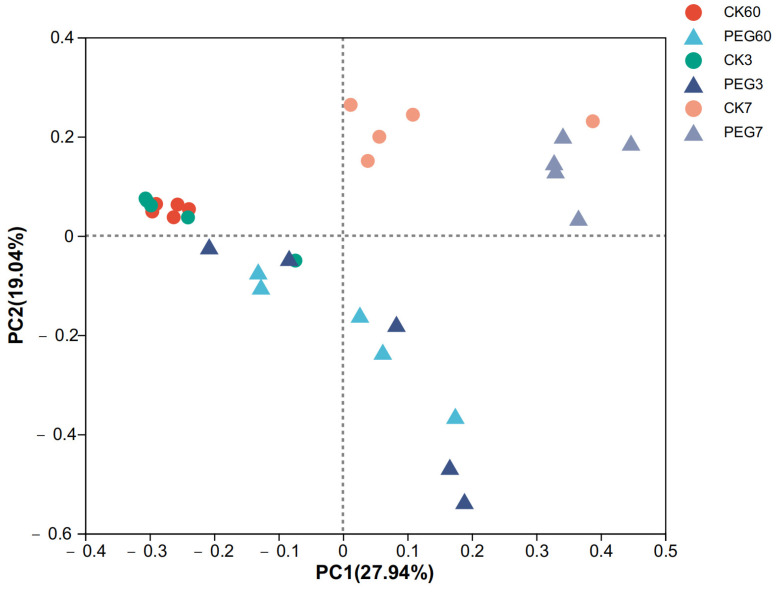
Principal coordinate analysis (PCoA) plot based on the weighted UniFrac distance for the fungal community of sainfoin silage. CK60: sainfoin ensiling after 60 d. CK3: sainfoin silage after 3 d of exposure. CK7: sainfoin silage after 7 d of exposure. CK: control; PEG: polyethylene glycol (R^2^ = 0.4715, *p* = 0.001).

**Figure 4 plants-12-02967-f004:**
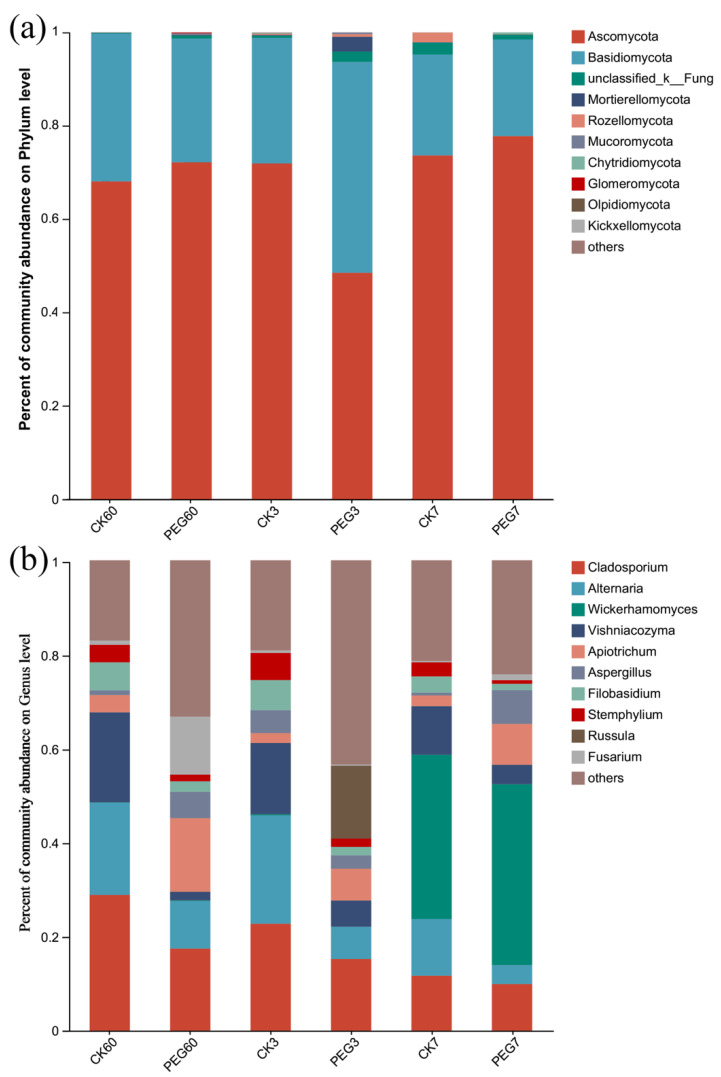
Fungal community at the (**a**) phylum and (**b**) genus level during the aerobic exposure of sainfoin silage. CK60: sainfoin ensiling after 60 d. CK3: sainfoin silage after 3 d of exposure. CK7: sainfoin silage after 7 d of exposure. CK: control; PEG: polyethylene glycol.

**Table 1 plants-12-02967-t001:** Characteristics of silage during ensiling and aerobic exposure.

Item	Treatment	Days of Ensiling	Days of Aerobic Exposure		SEM	*p* Value		
		60	3	7		Day	Treatment	D × T
DM	CK	24.42	24.81 B	24.78 B	0.8465	0.0146	<0.01	0.367
	PEG	25.42	26.40 A	26.30 A				
CP g/kg DM	CK	202.93 a	193.77 Ab	196.02 Ab	1.407	<0.01	<0.01	<0.01
	PEG	202.11 a	182.92 Bb	183.72 Bb				
pH	CK	4.55 Ab	4.62 Aa	4.66 Aa	0.011	<0.01	<0.01	<0.01
	PEG	4.55 A	4.54 B	4.54 B				
WSC g/kg DM	CK	24.33 Aa	11.80 b	10.53 b	1.137	<0.01	<0.01	<0.01
	PEG	11.24 B	7.43	5.26				
AN g/kg DM	CK	0.67 Ba	0.71 Ba	0.32 Ab	0.0403	<0.01	<0.01	<0.01
	PEG	0.77 Ab	0.94 Aa	0.30 Ac				
LA g/kg DM	CK	23.21 a	3.59 b	3.66 b	1.567	<0.01	0.132	0.08
	PEG	23.67 a	5.75 b	3.19 b				
AA g/kg DM	CK	16.49 a	11.41 b	13.34 b	0.412	<0.01	0.93	0.457
	PEG	16.14 a	12.33 b	12.63 b				
PA g/kg DM	CK	ND	ND	ND	-	-	-	-
	PEG	ND	ND	ND				
BA g/kg DM	CK	ND	ND	ND	1.153	0.132	<0.01	0.132
	PEG	14.90	12.38	10.88				
Yeast Log10 CFU/g FM	CK	2.98 Bb	3.91a	4.32 Aa	0.125	0.01	0.05	<0.01
	PEG	3.96 A	4.16	4.08 B				
LAB Log10 CFU/g FM	CK	6.48 Bb	6.77 Bb	6.95 Aa	0.1202	0.787	0.292	0.242
	PEG	7.15 Aa	7.20 Aa	6.63 Ab				
Mold Log10 CFU/g FM	CK	2.73 Bb	3.08 Aa	3.16 Aa	0.064	<0.01	0.029	<0.01
	PEG	3.41 Aa	2.68 Bb	2.42 Bb				
AB Log10 CFU/g FM	CK	6.09 b	6.88 b	7.19 a	0.106	0.002	0.937	0.088
	PEG	6.41	7.16	6.64				
Aerobic stability (h)	CK PEG	156.32 b 182.43 a			3.74	-	0.029	-

*n* = 3. CK: control; PEG: polyethylene glycol; DM: dry matter; CP: crude protein; WSC: water soluble carbohydrates; FM: fresh matter basis; AN: ammonia nitrogen; LA: lactic acid; AA: acetic acid; PA: propionic acid; BA: butyric acid; LAB: lactic acid bacteria; AB: aerobic bacteria; ND: not detected; SEM: standard error of the mean. A, B Means in the same column followed by different uppercase letters differ (*p* < 0.05); a–c means in the same row followed by different lowercase letters differ (*p* < 0.05).

**Table 2 plants-12-02967-t002:** Alpha diversity of bacteria after 60 d of ensiling and aerobic exposure in sainfoin silage ^1^.

Days	Treatment	Ace	Chao	Shannon	Simpson	Sobs	Coverage
60 d of ensiling	CK	121.03	114.60	1.78	0.2967	100.00	0.9995
	PEG	111.13	102.75	1.74	0.3224	84.60	0.9996
3 d aerobic exposure	CK	136.99	118.14	1.95	0.2403	90.00	0.9994
	PEG	113.79	92.52	1.70	0.3290	74.40	0.9994
7 d aerobic exposure	CK	149.12	140.47	2.10	0.2958	130.40	0.9994
	PEG	124.49	103.31	1.95	0.2416	82.40	0.9995
SEM		9.73	9.57	0.075	0.016	9.83	-
*p* value	Treatment (T)	0.72	0.78	0.37	0.59	0.63	-
	Day (D)	0.36	0.23	0.36	0.54	0.21	-
	T × D	0.95	0.88	0.86	0.23	0.75	-

^1^ *n* = 5. CK: control group; PEG: polyethylene glycol.

**Table 3 plants-12-02967-t003:** Alpha diversity of fungi after 60 d of ensiling and aerobic exposure in sainfoin silage ^1^.

Days	Treatment	Ace	Chao	Shannon	Simpson	Sobs	Coverage
60 d of ensiling	CK	189.28 a	159.31 a	3.00	0.1080	146.6 a	0.9995
	PEG	150.36	146.29 a	2.91	0.1067	134.8 a	0.9996
3 d aerobic exposure	CK	124.47	119.00	2.56	0.2001	110.6	0.9994
	PEG	105.64	104.33	3.03	0.1351	102.0	0.9994
7 d aerobic exposure	CK	91.63 b	83.55 b	2.91	0.1479	77.0 b	0.9994
	PEG	99.13	78.89 b	2.68	0.1467	65.0 b	0.9995
SEM		9.28	7.937	0.098	0.02	7.62	-
*p* value	Treatment (T)	<0.01	<0.01	0.75	0.53	<0.01	-
	Day (D)	0.274	0.396	0.82	0.62	0.37	-
	T × D	0.458	0.940	0.35	0.79	0.99	-

^1^ *n* = 5. CK: control group; PEG: polyethylene glycol. a, b: Means in the same column followed by different lowercase letters differ (*p* < 0.05) for the effect of days.

## Data Availability

The datasets supporting the conclusions of this article are available in the National Center for Biotechnology (NCBI) Sequence Read Archive (SRA) under accession numbers “PRJNA948115” and “PRJNA953494” for bacteria and fungi, respectively. http://www.ncbi.nlm.nih.gov (accessed on 1 August 2023).

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
