# Peer review of "Effects of Condensed Tannins on Bacterial and Fungal Communities during Aerobic Exposure of Sainfoin Silage"

_plants, 2023, doi:10.3390/plants12162967_

Round 1

Reviewer 1 Report

The Introduction is suggested to be concise to come straight to the point, and several separated paragraphs and more updated review literatures are needed.

I recommend expanding: Introduction, conclusions, and the discussion sections. The discussion still needs improvement and explanation in detail.

All the figures need to be revised with higher resolutions and also should be created in a consistent layout and style to improve the readability. Text in Figures are often too small to see.

The Tables should be checked for consistent formats and expression errors, including units, spaces between values and units, etc.

Transfer conclusion to the end of discussion

Please ensure that every reference cited in the text is also present in the reference list (and vice versa).

English should be improved; grammar needs enhancement in many sentences and paragraphs.

Author Response

Thanks for you advice, we carefully checked our manuscript, corrected some errors in our manuscript.

Reviewer 2 Report

See attached file.

Author Response

Thanks for you advice.

Reviewer 3 Report

The researchers conducted a study on the effects of condensed tannins on sainfoin silage and microbiological communities using various techniques such as NSG. Carry out a very complete study, in which important information is obtained. However, the authors could discuss more about the importance that the species of bacteria and fungi present in the silage could have. As well as the positive or negative impact that AA's ability to reduce the microbial community could have.

Below I list some details that the authors could modify.

Line 71, 74: it is the first time that the acronym BA, LA, AA is mentioned and it is not specified what the acronyms mean, add

Table 1: several acronyms are missing in the table footer

Figure 4 Names of microorganisms in italics in the graphs

the document requires minor revision in English

Author Response

Thanks for you advice. We carefully corrected some errors in our manuscript.

Line 71,74, Table 1: we added acronyms mean.

Figure 4: microorganisms in italics